# Photoregulated fluxional fluorophores for live-cell super-resolution microscopy with no apparent photobleaching

Elias A. Halabi[1], Dorothea Pinotsi[2] & Pablo Rivera-Fuentes [1]

Photoswitchable molecules have multiple applications in the physical and life sciences because their properties can be modulated with light. Fluxional molecules, which undergo rapid degenerate rearrangements in the electronic ground state, also exhibit switching behavior. The stochastic nature of fluxional switching, however, has hampered its application in the development of functional molecules and materials. Here we combine photoswitching and fluxionality to develop a fluorophore that enables very long (>30 min) time-lapse single-molecule localization microscopy in living cells with minimal phototoxicity and no apparent photobleaching. These long time-lapse experiments allow us to track intracellular organelles with unprecedented spatiotemporal resolution, revealing new information of the three-dimensional compartmentalization of synaptic vesicle trafficking in live human neurons.

[1] Laboratory of Organic Chemistry, ETH Zurich, Zurich 8093, Switzerland. [2] Scientific Center for Optical and Electron Microscopy, ETH Zurich, Zurich 8093, Switzerland. Correspondence and requests for materials should be addressed to P.R-F. (email: pablo.rivera-fuentes@org.chem.ethz.ch)

Substances that isomerize upon photoirradiation are useful as molecular devices because their properties can be switched at will with light[1–6]. Fluxional molecules, which undergo rapid degenerate rearrangements in the electronic ground state, exhibit stochastic switching behavior. Despite the great interest in this phenomenon in organic[7,8], coordination[9], main group[10], organometallic[11,12], and theoretical chemistry[13,14], only a few applications of fluxional molecules are known[15,16]. We envision that the juxtaposition of photoswitching and fluxionality could bring about new functional molecules with practical applications. Specifically, we hypothesize that such a molecule could be useful for single-molecule localization microscopy (SMLM) in living cells because it could ameliorate both phototoxicity and photobleaching, which are two critical limitations of this technique.

Most SMLM experiments rely on photoactivation of dyes or proteins[17–23]. Starting from a dark isomer, photoirradiation transforms a random subset of molecules to their fluorescent state. The emission signals of these sparsely distributed fluorescent molecules can be resolved individually, enabling the localization of single molecules (Fig. 1a). This approach has limitations in live-cell imaging because repeated photoactivation with light of high energy induces phototoxicity in the specimen and exacerbates photobleaching of the fluorophores (Fig. 1a). Whereas some implementations of SMLM achieve switching with a single wavelength[24], they still rely on relatively high, and therefore toxic, irradiation intensities. Other techniques, such as points accumulation for imaging in nanoscale topography, do not require photoactivation steps[25], but live-cell imaging of intracellular compartments remains very challenging[26].

To overcome these difficulties, spontaneously blinking dyes were developed[23,27,28]. These fluorophores exhibit a ground-state equilibrium between a fluorescent and a dark species, providing sparse distribution of fluorescent molecules without photoactivation, which greatly decreases phototoxicity and photobleaching (Fig. 1b). Despite the great potential of spontaneously blinking dyes, their performance in terms of resolution achieved, image quality, and apparent photobleaching depends on the fraction of molecules that are fluorescent at equilibrium. This fraction is strongly determined by the pH and polarity of the medium. Although these dyes have been used successfully to image specific molecular targets for short periods of time[28,29], long time-lapse imaging has only been realized in the low-polarity environment of membranes (Fig. 1b)[27]. In contrast, we argue that a combination of photoactivation and fluxionality would provide a way to control the fraction of fluorescent molecules independently from the properties of the medium. Starting from a dark, nonfluxional isomer, photoactivation would convert a fraction of the total molecules to a fluxional form (Fig. 1c). In this population of fluxional molecules, some would exist in a dark form and others in a fluorescent form. Regardless of what the fraction of fluorescent molecules is in the fluxional equilibrium, the total fraction of fluorescent molecules could always be controlled by photoactivation. Once a population of fluxional molecules is established, their thermal equilibrium between fluorescent and dark species could be used for single-molecule imaging with very low photoxicity. Moreover, even after the whole population of fluxional molecules is photobleached, a new subset of molecules could be photoconverted to the fluxional state (Fig. 1c), enabling extremely long time-lapse, single-molecule acquisitions with minimal phototoxicity.

Here, we report the design, synthesis, validation, and application of fluorescent molecules that become fluxional upon photoactivation. These processes are characterized by single-molecule

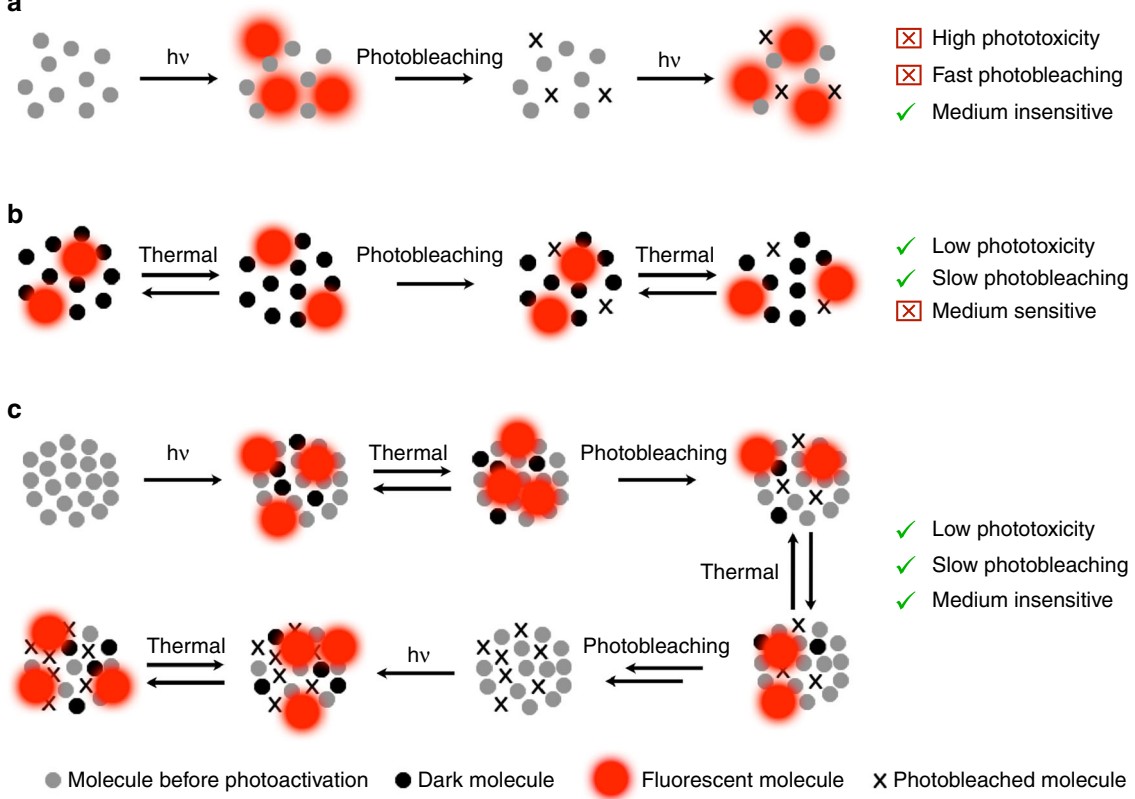

**Fig. 1** Mechanisms of single-molecule localization. **a** Classical mechanism employing photoactivatable dyes. **b** Spontaneously blinking dyes in a low-polarity environment (e.g., membranes). **c** Photoregulated fluxional fluorophores reported in this work

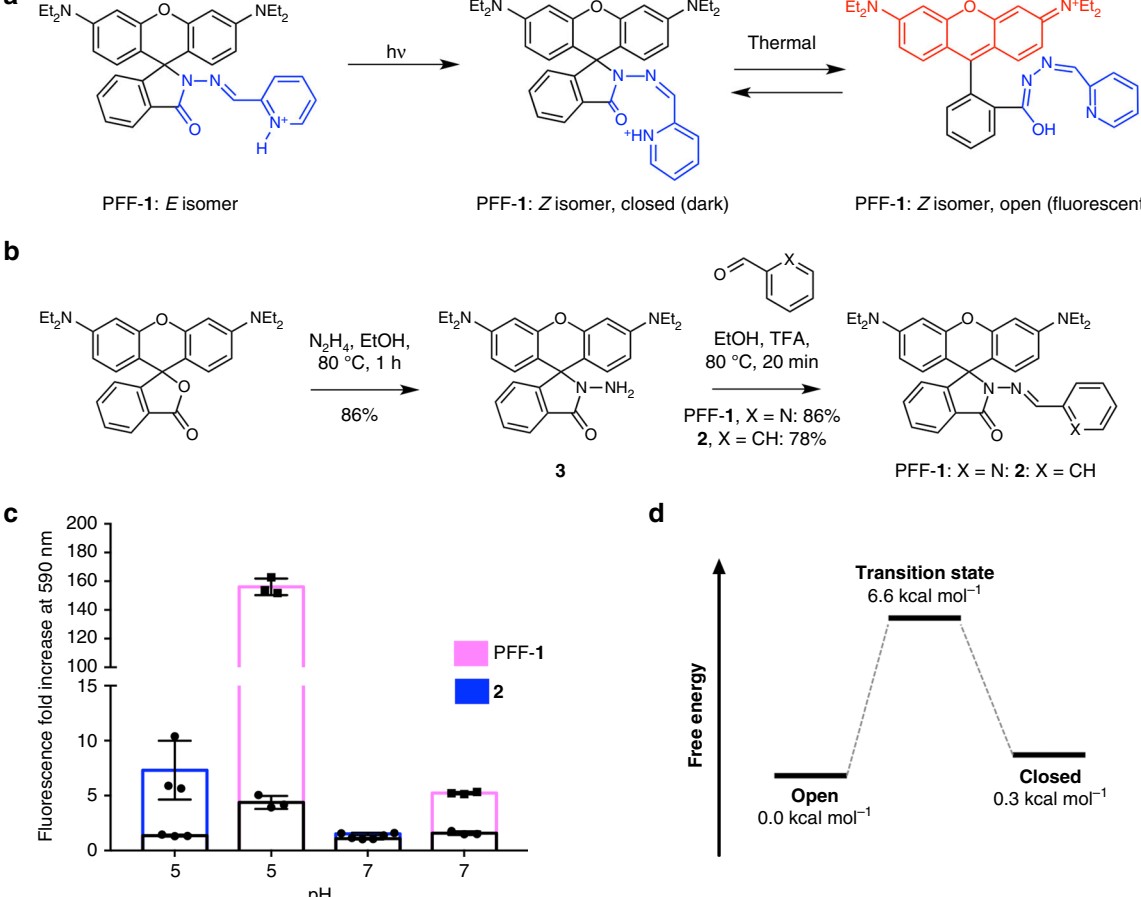

**Fig. 2** Synthesis and mechanism of probe PFF-**1**. **a** Mechanism of photoactivation and fluxionality of PFF-**1**. **b** Synthesis of probes PFF-**1** and **2** through their common intermediate **3**. **c** Fluorescence increase of compounds PFF-**1** and **2** upon photoirradiation at 410 nm in buffered aqueous solutions. Bars represent average values of three independent measurements ($N = 3$) and error bars indicate standard deviation (±). **d** Calculated potential energy surface for ring opening of the $Z$ isomer of probe PFF-**1**. The structures of "open" and "closed" correspond to those in panel (**a**). Relative energies were obtained at the B3LYP/def2-TZVP/IEFPCM($H_2O$) level of theory

imaging, demonstrating that the compound indeed becomes fluxional upon photoisomerization. Employing this probe, we are able to perform very long (>30 min) live-cell time-lapse SMLM, in 2D and 3D, with minimal toxicity and no apparent photobleaching. Finally, we apply this fluorophore to study the dynamics and three-dimensional compartmentalization of synaptic vesicle trafficking in live human neurons.

## Results

**Design, synthesis, and mechanistic studies**. To create photoregulated fluxional fluorophore PFF-**1** (Fig. 2a), we grafted an acylhydrazone photoswitchable unit[30,31] onto a rhodamine B scaffold. Probe PFF-**1** was synthesized in two steps and high yields from rhodamine B (Fig. 2b). As a mechanistic control compound, we also synthesized probe **2**, which could be obtained from intermediate **3** (Fig. 2b). We hypothesized that prior to photoactivation, PFF-**1** should exist predominantly as the $E$ isomer of the acylhydrazone and a dark, nonfluxional, spirocyclized derivative (Fig. 2a). This structure was confirmed by X-ray crystallography (Supplementary Fig. 1) and electronic absorption spectroscopy (Supplementary Fig. 2). Upon photoisomerization with light of 410 nm, the fluorescence of a solution of PFF-**1** in aqueous buffer (pH = 7) increases by 12-fold, whereas photoirradiation of compound **2** gives only a 1.6-fold increase under the same conditions (Fig. 2c). Furthermore, photoactivation of

PFF-**1** at pH = 5 led to an even larger increase in fluorescence of 22-fold, but for compound **2** the increase was only 3-fold.

$^1$H nuclear magnetic resonance (NMR) and high-performance liquid chromatography (HPLC) analysis of PFF-**1** prior to photoactivation confirmed that, in solution, virtually all molecules are present as the $E$ isomer (Supplementary Fig. 3). Upon irradiation in a ultraviolet (UV) photoreactor (10 min, 375 nm) a photostationary state was reached with about 17% conversion of the $E$ isomer into the $Z$ form, as determined by both $^1$H NMR and HPLC analysis (Supplementary Fig. 3). Photoisomerization using a light-emitting diode (LED) source (410 nm, 8.5 mW cm$^{-1}$) proceeded with a quantum yield of 0.4(2)% at pH = 5 and 0.5 (5)% at pH = 7.4 (Supplementary Figs. 4–6), confirming that the photochemical process does not depend strongly on pH. After photoirradiation, the $Z$ isomer did not revert to the $E$ form thermally over a period of 2 h (Supplementary Fig. 7).

Both the spirocyclic and open rhodamine forms absorb light at 254 nm with essentially the same extinction coefficient (Supplementary Fig. 2). In contrast, the open rhodamine form absorbs strongly at 560 nm, whereas the spirocyclic form does not absorb in this region at all. We reasoned that we could use the HPL chromatograms measured at these wavelengths to determine the fraction of molecules in both the $E$ and $Z$ isomers that are present as the open rhodamine form. Evaluation of these chromatograms revealed that the $Z$ isomer has a much higher absorbance at 560 nm compared to the $E$ isomer. This result indicates that

formation of the fluorescent rhodamine form is preferred in the $Z$, but not $E$ isomer. Based on these absorbance measurements (Supplementary Fig. 8), we determined that the percentage of molecules that is in the fluorescent state is approximately 39% for the $Z$ isomer, but only 0.004% for the $E$ isomer at pH = 5 (Supplementary Fig. 8). Because the population of the $Z$ isomer before irradiation is only 0.18% (Supplementary Fig. 8), the total percentage of fluorescent molecules is ~0.07%. After photoactivation (100 s, ~8.5 mW cm$^{-2}$), the population of the $Z$ isomer increases to 0.64%, which corresponds to ~0.25% of fluorescent molecules. These experiments confirm that even after photoactivation, the total fraction of fluorescent molecules is very small, which is a desirable feature for SMLM. Unfortunately, it was not possible to separate the two isomers at pH = 7.4 (Supplementary Fig. 5), but the overall fraction of fluorescent molecules could be estimated before and after photoactivation as 0.003% and 0.012%, respectively.

These experiments support the mechanism depicted in Fig. 2a, in which formation of the $Z$ isomer of the acylhydrazone in PFF-**1** brings the protonated 2-pyridyl substituent in close proximity to the carbonyl group, facilitating proton transfer-induced ring opening. This mechanism is further supported by density functional theory (DFT) modeling (Fig. 2d).

DFT modeling also suggests that the barrier of interconversion between the open fluorescent rhodamine and the dark spirocyclic form is energetically low (6.6 kcal mol$^{-1}$) for the $Z$ isomer (Fig. 2d). To demonstrate that the $Z$ isomer interconverts rapidly between the dark spirocyclic and fluorescent rhodamine forms

(i.e., it is fluxional), we embedded PFF-**1** in a polyvinylalcohol (PVA) film on a coverslip, irradiated this film with a photoactivation pulse (405 nm, 2.6 W cm$^{-2}$, 20 ms), and imaged single molecules with a 561 nm (0.25 kW cm$^{-2}$) laser in total-internal reflection mode. This method has been used before to evaluate the blinking properties of photoactivatable small molecules[32]. We validated the method by employing the nonfluxional compounds **2** and rhodamine B as negative controls and a spontaneously blinking molecule (HMSiR) as a positive control (Fig. 3 and Supplementary Fig. 9). This experiment revealed that single molecules of PFF-**1** transitioned several times between their dark and emissive forms, confirming the fluxionality of the probe (Fig. 3a, b). The switching behavior of PFF-**1** was robust, providing numerous switching cycles (221 ± 17, $N$ = 30) and a large number of emitted photons per switching cycle (640 ± 93, $N$ = 30). These parameters are comparable to those of commonly used SMLM dyes[32], but unlike these previously reported compounds, PFF-**1** does not require continuous photoactivation to switch. Moreover, single molecules of PFF-**1** could be localized with an average precision of 22 ± 9 nm in films, demonstrating that the fluxionality of PFF-**1** could be used for single-molecule localization.

**Time-lapse SMLM in living cells for more than 30 min**. The toxicity of PFF-**1** was assessed in HeLa cells (Supplementary Fig. 10). No loss of viability was detected at concentrations up to 100 μM (limit of solubility of PFF-**1** in growth medium) after

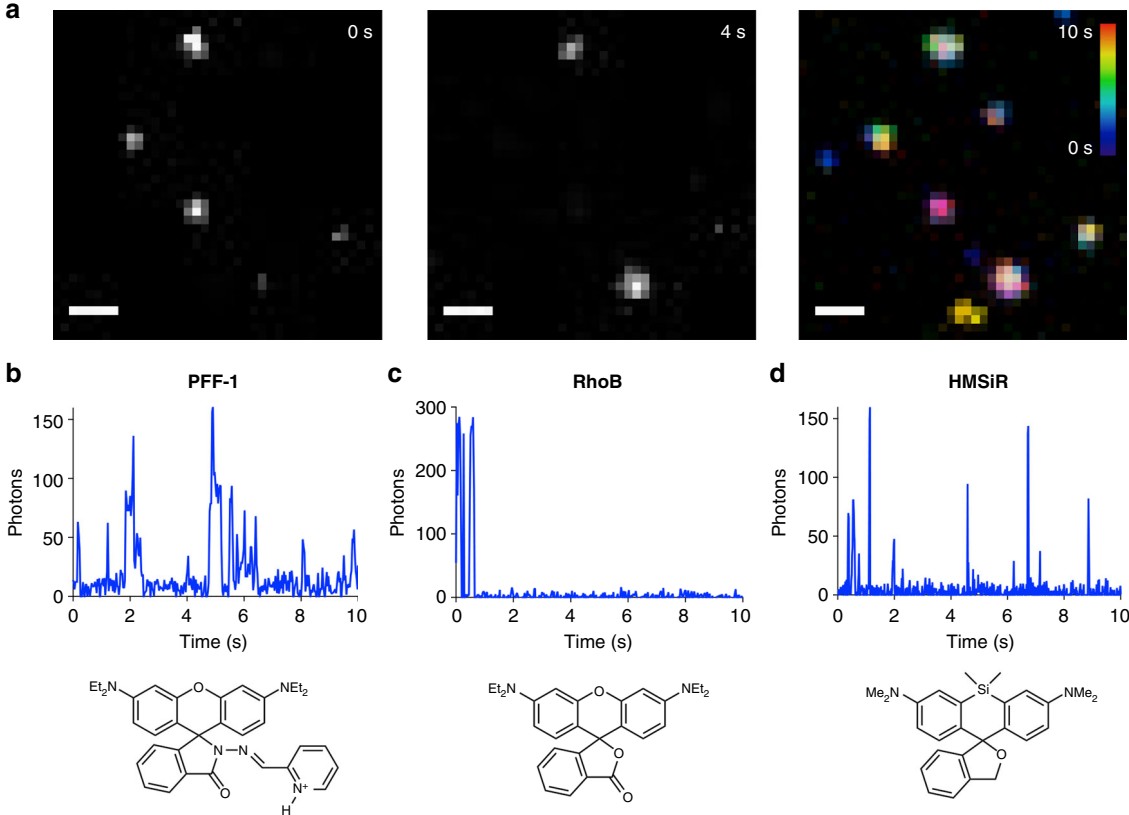

**Fig. 3** Single-molecule imaging of PFF-**1**, rhodamine B (Rho B), and HMSiR in polymer films. **a** Examples of camera frames in which single molecules of PFF-**1** were observed in the fluorescent state, and average intensity over all frames collected during a 10 s acquisition, color-coded according to time. Most spots in the color-coded image display multiple colors because single molecules emit photons at various time points. **b** Kymograph of a single molecule of probe PFF-**1** switching for 10 s. **c** Kymograph of a single molecule of Rho B. No switching is observed, only irreversible transformation into a dark state. **d** Kymograph of a single molecule of probe HMSiR blinking spontaneously for 10 s. The chemical structure of each compound is displayed under the indicated panel. Scale bar in panel (**a**) 1 μm

48 h. Spinning-disk confocal microscopy was used to prove that PFF-**1** diffused freely across the plasma membrane of mammalian cells and could be photoactivated with light of 405 nm (Supplementary Fig. 11). PFF-**1** displayed strong fluorescence after photoactivation in the acidic medium of lysosomes (Supplementary Fig. 12). Under identical photoactivation conditions, the fraction of fluorescent molecules of PFF-**1** is larger at low pH than in neutral solution because more molecules of PFF-**1** are protonated, which enables fluxionality. The total fraction of fluorescent molecules can be modulated by photoactivation and, unlike the spontaneously blinking dye HMSiR[28], PFF-**1** can be employed at either neutral or low pH, simply by tuning the photoactivation step.

We optimized the imaging conditions to record long time-lapse SMLM experiments in live HeLa cells using PFF-**1**. Prior to imaging, cells were incubated with 500 nM PFF-**1** for 10 min and then subjected to a photoactivation pulse (405 nm, 2.6 W cm$^{-2}$, 10 ms). After this initial photoactivation step, cells were either imaged using only the fluxionality of PFF-**1** (single photoactivation, Fig. 4a, c) or by repeating the photoactivation pulse every 10 min to replenish the population of fluxional molecules (sequential photoactivation, Fig. 4b, c). In either case, single molecules could be localized with an average precision of 33 nm over an acquisition time of 30 min. In the case of single photoactivation, however, substantial photobleaching occurred after 15 min (Fig. 4a, c). Sequential photoactivation (every 10 min), on the other hand, maintained the number of detected molecules essentially constant during the whole acquisition (Fig. 4b, c and Supplementary Movie 1). This experiment confirms that PFF-**1** enables extremely long time-lapse SMLM at irradiation intensities that are much lower than those used in typical experiments (Supplementary Table 1). Furthermore, we evaluated the morphology of the cells, disruption of the plasma membrane, and activation of caspase-3 as signs of phototoxicity. Even after this very long time-lapse acquisition, the irradiated cells displayed only modest signs of phototoxicity (Supplementary Fig. 13), highlighting the advantages of using a combination of very mild photoactivation and fluxionality for single-molecule localization.

Under these imaging conditions, we were able to distinguish individual vesicles within diffraction-limited areas (Supplementary Fig. 14), confirming that cellular structures that would have been blurred by diffraction became discernible using our imaging approach. Moreover, the overall high-labeling density and fast switching kinetics of PFF-**1** allowed us to localize approximately 1100 molecules per μm$^2$ within vesicles using only 23 camera frames. Using these signals, we were able to obtain reconstructed snapshots with a Nyquist-limited resolution of nearly 60 nm every 0.5 s. This excellent spatiotemporal resolution allowed us to track in detail the fast motion of single lysosomes. The example displayed in Fig. 4d depicts a vesicle displaying both fast directional motion and slow Brownian-like diffusion, in agreement with previous reports of lysosome dynamics[33].

We also performed time-lapse 3D SMLM using astigmatism and adaptive optics[34]. These images revealed the 3D distribution of lysosomes within a 1 μm-thick imaging volume in a living cell (Fig. 4e, f). The adaptive optics implementation allowed us to increase the localization precision to $11 \pm 5$ and $140 \pm 60$ nm in the lateral and axial direction, respectively. A single lysosome moving primarily along the axial direction could be tracked in this experiment (Fig. 4g), illustrating the level of spatiotemporal detail with which these organelles could be visualized employing probe PFF-**1**.

Besides being useful to image and track lysosomes, probe PFF-**1** could also be used to label other cellular components at higher pH. We synthesized probe MitoPFF-**1**, which is a derivative of PFF-**1** functionalized with a mitochondria-targeting triphenylphosphonium group[35]. Although mitochondria display neutral or slightly basic pH, and therefore would decrease the fluorescent fraction of MitoPFF-**1** in its fluxional equilibrium, we were able to image mitochondria with super-resolution using this probe simply by tuning the photoactivation conditions (Supplementary Fig. 15). Additionally, a taxol conjugate of PFF-**1**, termed TaxoPFF-**1**, was prepared to test whether PFF-**1** remains fluxional upon binding to a large macromolecular target such as microtubules. Super-resolved images of single microtubules were reconstructed from fixed HeLa cells after 120 s of acquisition. Under these conditions, single microtubules could be resolved with a full width at half maximum (FWHM) of $100 \pm 10$ nm (Supplementary Fig. 16). Although further optimization of the labeling strategy and image acquisition could improve this resolution, these experiments already demonstrate that photo-regulated fluxional fluorophores could be used to image a variety of targets by optimizing the irradiation pulses, labeling and imaging conditions.

**Tracking synaptic vesicles in live neurons**. Having shown that probe PFF-**1** enables single-vesicle tracking, in three-dimensions, and with subsecond time resolution, we set out to image synaptic vesicles in human neurons (Fig. 5a and Supplementary Fig. 17) derived from neuroblastoma cells[36]. We first confirmed that PFF-**1** targeted synaptic vesicles by co-localization analysis with FM1–43, a common stain for these vesicles (Supplementary Fig. 18). Using PFF-**1**, we could resolve two vesicles within a diffraction-limited spot ($250 \times 250$ nm), corroborating that these fast-moving organelles could be resolved beyond the limit of diffraction (Supplementary Fig. 14). Having established that we could image synaptic vesicles with excellent spatiotemporal resolution, we analyzed their movements. We found that these vesicles exhibited a very broad range of diffusion coefficients, ranging from very static ($V_1$, $D = 0.0005$ μm$^2$ s$^{-1}$, Fig. 5b) to extremely fast ($V_2$, $D = 3.206$ μm$^2$ s$^{-1}$, Fig. 5b and Supplementary Movie 2). Time-lapse SMLM confirmed that slow vesicles accumulate in hotspots and fast vesicles move along tracks, in agreement with previous reports[37]. Our long time-lapse experiments, however, allowed us to discover that vesicles travel along axons by hopping from one hotspot to another within a few seconds, often with long residence times in hotspots. This behavior is exemplified by vesicle $V_3$, which diffuses between hotspots $H_1$ and $H_2$ (Fig. 5c and Supplementary Movie 3). This hotspot hopping mechanism provides neurons with a constant pool of vesicles localized in defined compartments (hotspots), but also with the ability to exchange vesicles rapidly between them through tracks. Intrigued by this observation, we performed 3D SMLM at an axonal synapse (Fig. 5d) to uncover additional information of how vesicles move along tracks and join hotspots. These experiments revealed that tracks ($T_1$ and $T_2$, Fig. 5e, f) are localized approximately 1 μm higher (relative to the coverslip) along the axial direction than hotspots $H_3$–$H_6$. These tracks, however, have connections to these hotspots (Fig. 5f). The diffusion of a vesicle between $H_3$ and $H_4$ confirmed that vesicles are able to jump out of the hotspot, join the track, move rapidly through it, and fall into the next hotspot (Fig. 5g). A reconstructed volume over the whole acquisition (Supplementary Movie 4) clearly illustrates the morphology of the hotspots, vesicle tracks, and how vesicles move within and between hotspots in three-dimensions. More examples of this phenomenon are displayed in Supplementary Fig. 19.

## Discussion

The combination of photoactivation and fluxionality in the same fluorophore allowed us to control its emission in a manner that alleviates photobleaching and phototoxicity in live-cell SMLM.

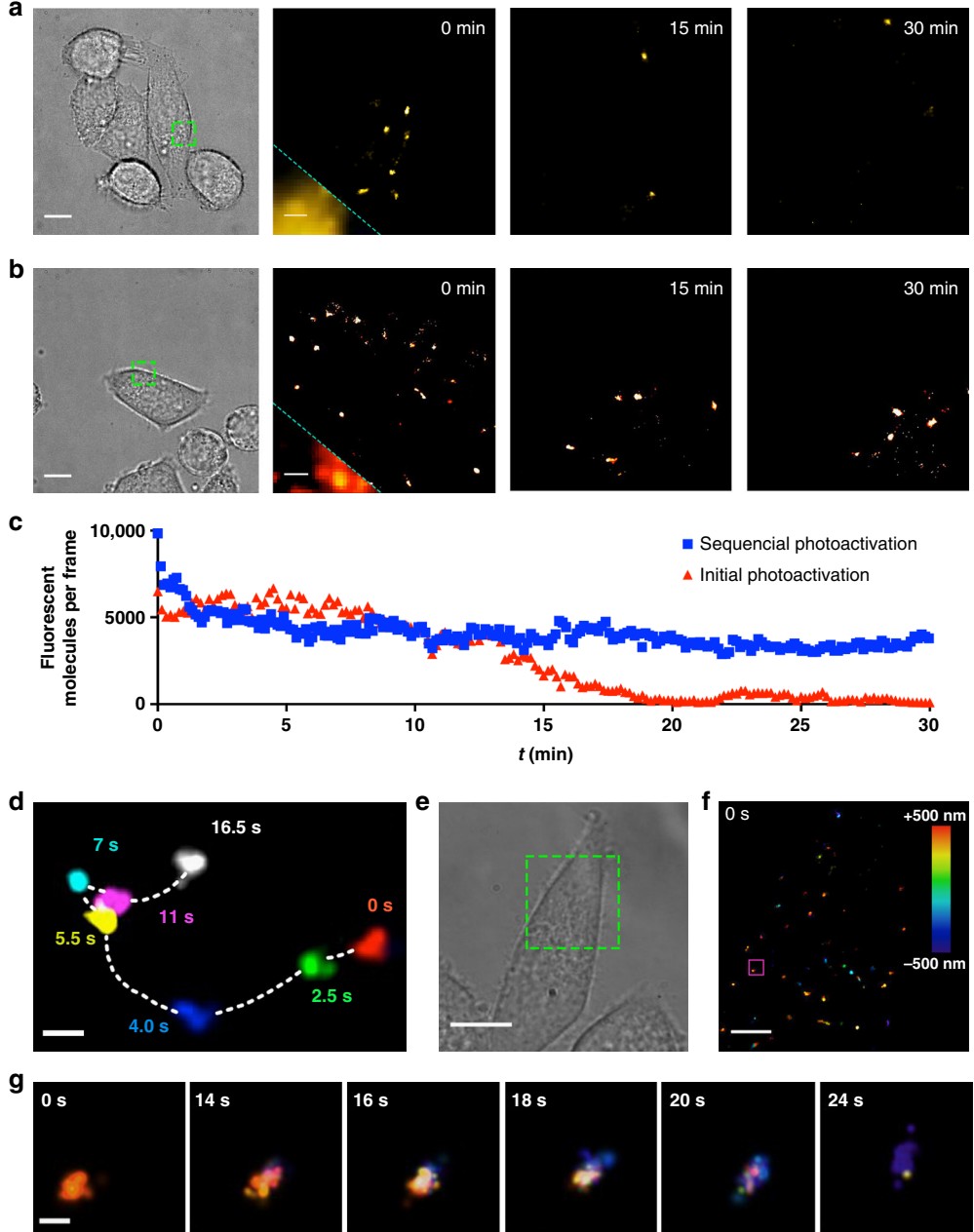

**Fig. 4** Time-lapse, 2D and 3D SMLM of lysosomes in live HeLa cells using PFF-**1**. **a** Time-lapse SMLM of lysosomes in a live HeLa cell applying a single photoactivation pulse (405 nm, 2.6 W cm$^{-2}$, 20 ms). The green, dotted square in the bright-field image indicates the area imaged during the SMLM experiment. The bottom left corner of the image at 0 s displays the diffraction-limited image for comparison. **b** Time-lapse SMLM of lysosomes in a live HeLa cell applying photoactivation pulses (405 nm, 2.6 W cm$^{-2}$, 20 ms) every 10 min. The green, dotted square in the bright-field image indicates the area imaged during the SMLM experiment. The bottom left corner of the image at 0 s displays the diffraction-limited image for comparison. **c** Number of emissive molecules per frame over the entire acquisition employing probe PFF-**1** (500 nM) with an initial, single-photoactivation pulse (405 nm, 2.6 W cm$^{-2}$, 20 ms, red triangles) or with the same pulse applied every 10 min (blue squares). **d** Single lysosome moving laterally within the focal plane in a live HeLa cell. **e** Bright-field image of a live HeLa cell. The green, dotted square indicates the area displayed in panel (**f**). **f** Snapshot at 0 s of a 3D, time-lapse, SMLM experiment, color-coded according to depth in a 1 μm-thick imaging volume. The magenta, solid-line square indicates the area displayed in panel (**g**). **g** Snapshots at different time points of a single lysosome diffusing in the axial direction, color-coded according to the scale depicted in panel (**f**). Scale bars from panels (**a**, **b**) 10 μm (bright-field) and 1 μm (diffraction-limited and SMLM), respectively; (**d**) 500 nm; (**e**) 10 μm; (**f**) 5 μm; (**g**) 200 nm

This feature facilitates very long (>30 min) time-lapse imaging of single-molecules within living cells, with a temporal resolution of seconds, even in small and densely labeled subcellular compartments. To the best of our knowledge, our time-lapse SMLM experiments are the longest ever reported, and even after such long acquisition times, no apparent photobleaching was observed. The performance of these dyes was not affected by the medium, allowing for super-resolved imaging of both neutral and acidic intracellular organelles, and thereby providing a generalizable mechanism for the development of small-molecule dyes for long-term SMLM.

Despite its many advantages, PFF-**1** also has a few potential shortcomings. For example, the fraction of Z isomers must be kept low to avoid overlap of multiple emitters. Because of this low

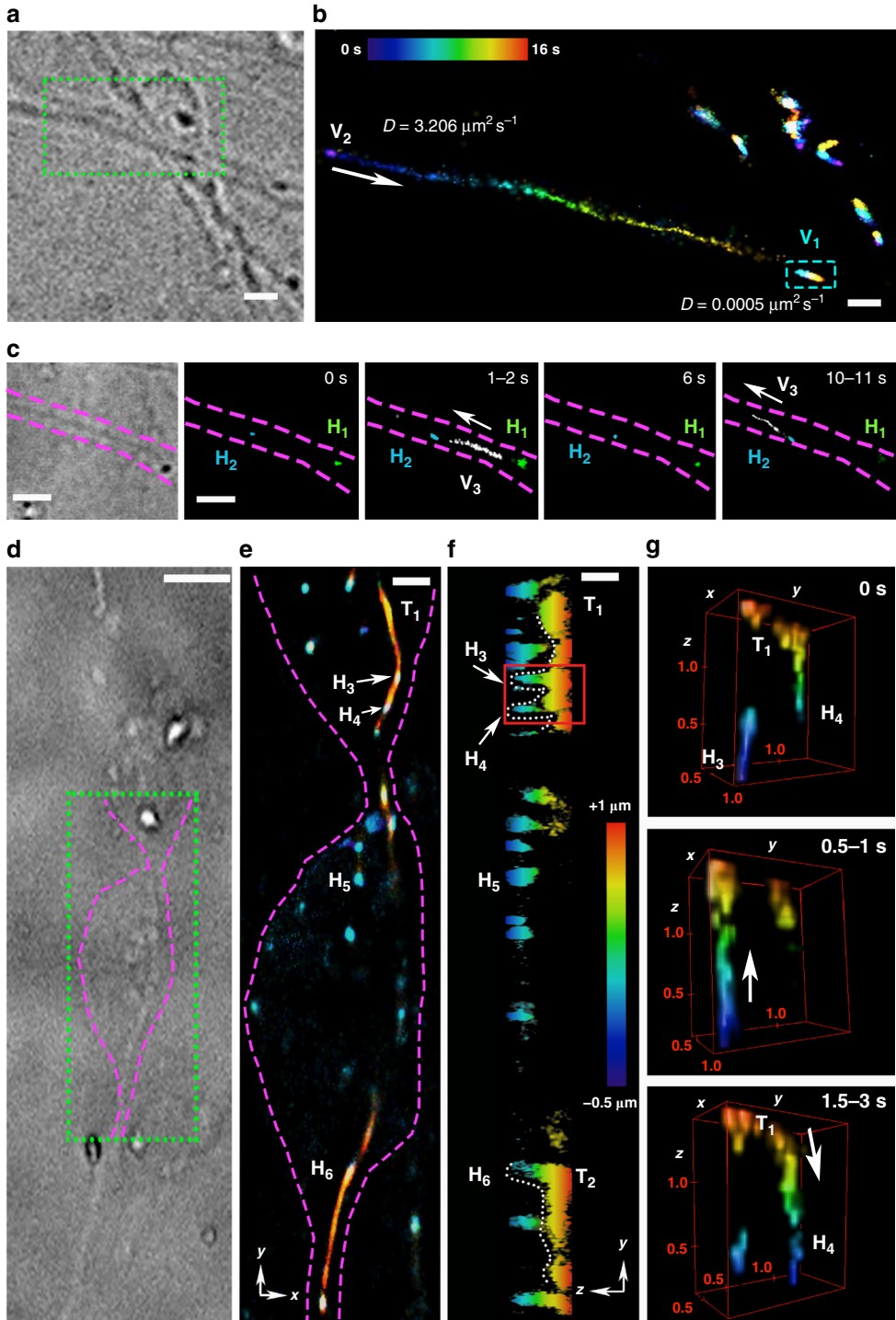

**Fig. 5** Single-vesicle tracking in human neurons. **a** Bright-field image of a neuronal projection and synapse. The green, dotted box indicates the area depicted in panel (**b**). **b** Image reconstructed from all frames collected during a 16 s acquisition and color-coded according to time. Diffusion coefficients were determined for two vesicles ($V_1$ and $V_2$). The cyan, dotted rectangle indicates the area in which $V_1$ moves. Other unlabeled vesicles are also present in the top-right area of the image. (**c**) Bright-field image of a single neuronal projection and fast diffusion of an acidic vesicle ($V_3$) between hotspots ($H_1$ and $H_2$) in 20 s. The arrows indicate the direction of travel of the fast-moving vesicle. **d** Bright-field image of two neuronal projections meeting at a synapse. The green, dotted box indicates the area depicted in panel (**e**). **e** 3D SMLM depiction merging all frames acquired over 40 s, giving a long-term image in which tracks of fast vesicle movement emerge and color-coded according to depth (see color bar in panel (**f**)). **f** Image of panel (**e**) projected across the axial plane (*y–z*), color-coded according to depth. A region of interest (ROI) containing track $T_1$ and hotspots $H_3$ and $H_4$ was selected (red, solid-line square) to display as a time series and 4D volume in panel (**g**). **g** 3D images at selected time points displaying a vesicle moving from hotspot $H_3$ to $H_4$ through track $T_1$. The volumes were colored according to the bar in panel (**f**). The unit cell is 1.5 μm × 1.5 μm × 1.5 μm and labeled with ticks of 0.5 μm. Scale bars from panels (**a**, **b**) 2 μm; (**c**, **e**) 1 μm, (**d**) 5 μm; (**f**) 750 nm

density of emitting molecules, it would be more difficult to resolve complex cellular structures that require a large number of localizations with high-temporal resolution. The development of new photoregulated fluxional fluorophores with increased brightness and blinking speed might ameliorate this situation.

Employing this dye and imaging methodology, a series of time-lapse 2D and 3D SMLM experiments in neurons revealed that synaptic vesicles move along well-defined tracks and accumulate within hotspots. Although tracks and hotspots have been identified before[37], our long time-lapse acquisitions revealed that vesicles hop between hotspots using tracks. We also determined that vesicles in tracks move more than three orders of magnitude faster than vesicles at hotspots. Moreover, time-lapse 3D SMLM demonstrated that tracks and hotspots occupy distinct locations along the axial direction of the cell, but they fuse to one another at defined points to enable the exchange of vesicles. These investigations confirm that the probes reported in this work are robust, generally applicable, and enable the discovery of biological mechanisms that would have been overlooked with other probes or imaging techniques.

## Methods

**Synthesis and general methods**. The synthesis and characterization of all compounds and additional details of experimental methods are available as Supplementary Methods.

**General methods**. All reagents were purchased from commercial sources and used without further purification. Solvents were obtained from Sigma-Aldrich. NMR spectra were acquired on a Bruker AV400 and AV500 instruments. $^1$H NMR and $^{13}$C NMR chemical shifts are reported in ppm relative to SiMe$_4$ ($\delta = 0$) and were referenced internally with respect to residual protons ($\delta = 7.26$ for CD$_3$Cl, $\delta = 3.31$ for CD$_3$OD and $\delta = 1.94$ for CD$_3$CN) and carbons ($\delta = 118.26$ for CD$_3$CN, $\delta = 77.16$ for CD$_3$Cl and $\delta = 49.00$ for CD$_3$OD) in the solvent respectively. Coupling constants are reported in Hz. Peak assignments are based on calculated chemical shifts, multiplicity and 2D experiments. IUPAC names of all compounds are provided and were determined using CS ChemBioDrawUltra 15. High-resolution mass spectra were recorded by staff at the molecular and bioanalytical (MoBiAS) center (ETH Zurich) employing a Bruker maXis-ESI-Qq-TOF-MS (electrospray ionization (ESI)).

**Fluorescence spectroscopy and photoactivation experiments**. Stock solutions in dimethylsulfoxide (DMSO) were prepared (1 mM), stored at −20 °C, and thawed before each experiment. UV-visible spectra were acquired employing a Cary 500 Scan spectrometer. Fluorescence spectra were acquired using a Micromax plate reader coupled to a Fluorolog 3 fluorimeter (Horiba Jobin-Yvon). All measurements were conducted at 25 °C and under red light ambient illumination to avoid photoactivation of the compounds. Microplate experiments with compounds PFF-**1** and **2** (100 μM, 100 μL) were performed in citric acid and Na$_2$HPO$_4$ buffers (pH = 3, 4, 4.5, 5, 6, 6.5, 7, 7.5, and 8) in a black-wall, clear bottom, polysterene 96-well plates (Corning Inc.). Cuvette experiments were performed in citric acid and Na$_2$HPO$_4$ buffers at pH = 5 and 7.4. All measurements were performed in triplicate. Photoactivation was achieved with a home-built LED transilluminator (410 nm, 2 mW cm$^{-2}$) for 10 min. Fluorescence intensity was measured as a single point (590 nm) before and after photoactivation. The data were plotted as the fluorescence intensity normalized to the initial intensity ($I = 1$).

**Quantum yield of photoactivation**. Stock solutions of PFF-**1** (500 μM) were prepared in CH$_3$CN and sonicated prior to the experiment. Citric acid:Na$_2$HPO$_4$ buffer solutions (pH = 5 or pH = 7.4, 5 mM citric acid and 10 mM Na$_2$HPO$_4$) were used to adapt the pH during irradiation and were verified with a pH meter (VWR), filtered and degassed before use. Diluted PFF-**1** solutions (50 μM) in CH$_3$CN:buffer (7:3) were prepared (3.2 mL) and irradiated in quartz cuvettes (Thorlabs) under constant magnetic stirring. An LED light source (405 nm, Roithner Lasertechnik, LED405-06V) was placed in direct contact with the cuvette containing the solution. A power-meter (Thorlabs, PMD100D) equipped with a Si photodiode detector (Thorlabs, S120VC) was used to measure the total photon flux. The transmitted light and the total effective absorbed power were monitored during each acquisition. The incident light measured with the blank solution was adjusted to ~8.5 mW before each irradiation. Aliquots of the solutions (200 μL) before and after irradiation were transferred into the HPLC autosampler and 100 μL were injected for analysis. Six time points were taken in intervals of 20 s ($t = 0$, 20, 40, 60, 80, and 100 s). An Excel 5 CN-ES UHPLC column (ACE®) was used to separate the compounds at 40 °C. All experiments were measured in triplicate. The chromatograms were exported as ASCII files and imported into Prism 7 (GraphPad). The

time range of the peaks of interest were selected, a minimum value of each curve was subtracted from all the values within the peak serving as a baseline correction. The area under the curve and linear fit of the areas were calculated using Prism 7 (GraphPad). For small areas with irregular baseline (Supplementary Fig. 8d, f) the total area was estimated as the difference between the total peak and the highlighted triangular section.

The concentration (μM) was determined from the initial concentration of the solution (50 μM) and the molar fraction of the $E$ and $Z$ isomers calculated from the areas under the peaks in the HPL chromatogram. The difference in concentration ($\Delta c$) was determined. The differences in molecules $\Delta M$ were determined by the product of the difference in concentration ($\Delta c$), the irradiated volume ($V_{\text{eff}}$), and Avogadro's number ($N_a$, Eq. (1)).

$$\Delta M = V_{\text{eff}} \times N_a, \tag{1}$$

The quantum yield of photoactivation ($\Phi_{\text{PA}}$) is given by the ratio of the differences in molecules and the number of absorbed photons per time interval $N_{\text{Photons}}$ (Eq. (2)).

$$\Phi_{PA} = \Delta M \div N_{\text{Photons}}, \tag{2}$$

$N_{\text{Photons}}$ can be evaluated from the measured absorbed power ($I_{\text{abs}}$) and the energy of a photon ($E_{\text{Photon}}$) at the irradiated wavelength (405 nm) in time ($\Delta t$) (Eq. (3)).

$$N_{\text{Photons}} = I_{\text{abs}} \times \Delta t \div E_{\text{Photon}}. \tag{3}$$

**Determination of ON states**. A calibration curve of rhodamine B monitored at 560 nm was built (0.05, 0.1, 0.25, 0.5, and 1 μM) to estimate the total number of ON molecules. The areas under the curve were obtained in triplicate from the HPL chromatograms of PFF-**1** measured after irradiation ($t = 100$ s) at pH 5. The slope and $y$-intercept obtained from this linear regression were used to estimate the concentration of photoproducts for both $E$ and $Z$ isomers. The ratio of the ON molecules monitored at 560 nm (0.123 and 0.002 μM; $Z$, $E$, respectively) and total molecules monitored at 254 nm (0.313 and 49.67 μM; $Z$, $E$, respectively) were given as percentages. The total number of molecules in the ON state was calculated from the ratio of the sum of ON $Z$ and $E$ molecules and the total concentration (0.25%).

**Single-crystal X-ray crystallography**. Clear, colorless single crystals of compound PFF-**1** C$_{34}$H$_{35}$N$_5$O$_2$ ($M = 545.67$ g mol$^{-1}$, CCDC 1844696, Supplementary Fig. 1) were obtained (approximate size $0.171 \times 0.091 \times 0.027$ mm) by slow evaporation of methanol at 25 °C. The monoclinic crystal system belongs to the space group P 2$_1$/n ($a = 16.9272(5)$ Å, $b = 25.6415(2)$ Å, $c = 9.3916(3)$ Å, $\beta = 136.319(5)°$, $V = 2815.3$ (2) Å$^3$, $Z = 4$, μ(CuKα) = 0.647 mm$^{-1}$, $Dc = 1.287$ g/cm$^3$). A total of 29,308 reflections were measured (6.894° ≤ 2Θ ≤ 159.776°) out of which 5991 unique ones were used in all calculations ($R_{\text{int}} = 0.0320$, $R_{\text{sigma}} = 0.0244$). Final 0.0447 ($I > 2\sigma$ ($I$)) and $wR_2$ was 0.1198.

**Computational modeling**. Structures of the $E$ and $Z$ isomers of compound PFF-**1** were optimized in the open- and closed-ring isomers at the B3LYP/def2-TVZP level of theory, employing an intrinsic solvation model (IEFPCM) for H$_2$O in Gaussian 09. The two minima were characterized by zero imaginary vibrational frequencies. The transition state was located by performing a relaxed scan of elongation of the C–N bond that forms the lactam ring (steps of 0.02 Å). The structure of highest energy in this relaxed potential energy surface was used as input for optimization of the transition state, which was characterized by one imaginary vibrational frequency involving the elongation of the C–N bond of interest. Energies of stationary states were corrected by their zero-point energy.

**Cell viability assay**. Viability of HeLa cells upon incubation with compound PFF-**1** was determined using the methylthiazolyldiphenyl-tetrazolium bromide (MTT) assay. Each well of a 96-well plate contained 9000 HeLa cells that were previously grown to 90% confluency. The cells were allowed to attach for 24 h in complete growth medium (10% fetal bovine serum, FBS). A solution of compound PFF-**1** (20 mM) was prepared in DMSO and diluted with growth medium so that the percentage of DMSO remained constant throughout the concentration range (0.5%). The cells were treated with compound PFF-**1** in different concentrations (100 μM, 50 μM, 25 μM, 12.5 μM, 6.25 μM, 3.13 μM, 1.56 μM, 781 nM, 390 nM, and 195 nM). The cells were incubated for 48 h at 37 °C under a humidified atmosphere containing 5% CO$_2$. To determine viability, the wells were treated with 10% MTT solution (5 g L$^{-1}$) in imaging medium (Fluorobrite) and incubated for 3 h. The supernatant was discarded and replaced with isopropanol (100 μL). The plates were shaken at 450 rpm in a microplate shaker (VWR) and absorbance was recorded for each well with a plate reader (SPARK 10 M, TECAN). Duplicates of every three independent experiments ($N = 6$) were measured for every concentration. A well with DMSO only (0.5%) was used as a positive control (100% viability). Absolute IC$_{50}$ values were determined with Prism 7 (GraphPad) and error bars represent standard deviations.

**Cell culture and confocal imaging**. Human cervical cancer HeLa cells were obtained from American Type Culture Collection (ATCC®-CCL-2$^{\text{TM}}$), grown in

Dulbecco's Modified Eagle Medium (DMEM) containing FBS (10%) and penicillin–streptomycin (0.1%) and incubated at 37 °C in a 95% humidity atmosphere containing 5% $CO_2$. The cells were grown to 90% confluence before plating on a Nunc$^{TM}$ Lab-Tek$^{TM}$ chambered cover glass at a density of 50,000 cells mL$^{-1}$ for 48 h. Prior to imaging, the medium was removed, the cells were washed with phosphate-buffered saline (PBS) (pH = 7.4) and imaged with the defined concentration of probe using phenol red-free imaging medium (Fluorobrite). Confocal microscopy was performed with a Nikon Eclipse T1 microscope equipped with a Yokogawa spinning-disk confocal scanner unit CSU-W1-T2, a LUDL BioPrecision2 stage and an sCMOS camera (Orca Flash 4.0 V2). An objective was used with a magnification of 100 × 1.49 CFI Apo TIRF using oil immersion. Light sources were diode-pumped, solid-state lasers (DPSS): 405 (120 mW), 445 (100 mW), and 561 (200 mW). Laser powers and exposure times were kept constant within specific experiments. The microscope was operated using VisiVIEW (Metamorph) software.

**Neuroblastoma culture, differentiation, and preparation for imaging**. Human-derived neuroblastoma SH-SY5Y cells were obtained from ATCC (ATCC®-CRL-2266), and cultured and differentiated following a reported protocol[36]. Differentiation into mature neurons was confirmed using a primary antibody against phosphorylated neurofilament H (SMI31, 801602, BioLegend) and mouse IgG (H + L) highly crossed-adsorbed secondary antibody (Alexa Fluor Plus 488, A32723, ThermoFisher). Neurons were fixed with a 4% formaldehyde solution in PBS for 15 min, permeabilized with 0.25% Triton X-100 in PBS for 10 min and blocked with 3% bovine serum albumin (BSA) in PBS for 30 min at 25 °C. Cells were treated with a solution (1:200) of the primary antibody in PBS with 3% BSA for 1 h at 25 °C, followed by incubation with the secondary antibody in a solution (1:1000) in PBS with 3% BSA for 2 h at 25 °C. Neurons were washed with PBS before imaging (488 nm, 0.18 kW cm$^{-2}$, 10 ms). For 2D and 3D SMLM imaging of live neurons, differentiated cells were washed with PBS once and treated with compound PFF-**1** (500 nM, 10 min) without any further washing step.

**Cloning and transfection for co-localization experiments**. HeLa cells were transiently transfected with plasmids encoding for the cyan fluorescent protein mTurquoise2 fused to targeting sequences for mitochondria (Addgene #36208), ER (Addgene #36204), and Golgi (Addgene #36205)[38]. The nucleus was stained with Hoechst 33342 (10 μM). Lysosomes were labeled by fusing mTurquoise2 to lysosome-associated membrane protein 1 (LAMP1). This plasmid was prepared by Gibson assembly[39] employing sequences from commercial plasmids pmTurquoise2-ER (Addgene #36204), from which the insert was taken, and pLAMP1-mCherry (Addgene 45147), which served as backbone, employing the following primers: insert forward (5′-TATGGTGAGCAAGGGCGAGGAG-3′), insert reverse (5′-ACAGCTCGTCCATGCCGAGAGT-3′), backbone forward (5′-GGCATGGACGAGCTGTACAAGGCTAC-3′) and backbone reverse (5′-TCGCCCTTGCTCACCATACCG-3′). The insert and backbone DNA fragments were generated by polymerase chain reaction (PCR) using Phusion High-Fidelity PCR Master Mix with HF buffer (NEB, M0531S). PCR products were purified using QiAquick PCR purification kit and analyzed by gel electrophoresis. The insert and backbone DNA fragments were assembled employing Gibson Assembly Master Mix (NEB, E3611). The assembled product was transformed into competent *E. coli* (DH5α) cells, plated onto lysogeny broth (LB) agar containing 100 μg mL$^{-1}$ carbenicillin and incubated at 37 °C for 24 h. DNA was extracted with the QiAprep spin miniprep kit and sequenced using a standard primer source CMV-for (5′-CGCAAATGGGCGGTAGGCGTG-3′) and BGH-rev (5′-TAGAAGGCA-CAGTCGAGG-3′). Mammalian cells were transfected after an incubation time of 24 h after plating and 48 h before confocal imaging using plasmid DNA (0.5 μg) and jetPRIME® according to the manufacturer instructions (Polyplus). The transfection reagent was allowed to react for 5 h and the medium was exchanged with fresh growth medium (10% FBS).

**Single-molecule imaging in PVA films**. Eight-well glass bottom chambers (Ibidi) were thoroughly cleaned by sonication (30 min) in NaOH (1 M) solution followed by etching for 2 h. The chamber was rinsed with abundant Mili-Q water and subsequently sonicated (30 min). The chamber was treated with UV-grade ethanol and allowed to dry at 50 °C. A polyvinylalcohol solution (0.1%) solution was prepared in PBS (pH′ = 7.4). Preparation method A: 400 μL of PVA (1%) were added to each well and left to evaporate at 50 °C overnight. Before imaging, 400 μL of a solution containing the compound (PFF-**1**, rhodamine B, or HMSiR; 1 nM) at the desired pH (5 for PFF-**1** or compound **2** or 7.4 for rhodamine B and HMSiR) was added, incubated for 5 min, washed thoroughly with the same buffer, and imaged using a perfect focus system (PFS) for SMLM. Preparation method B: 20 μL of PVA (1%) solution containing the desired compound (1 nM) were added to each well and left to evaporate at 50 °C overnight. Before imaging, 200 μL the appropriate buffer at pH (5 or 7.4) was added, washed, and imaged using the PFS setting for SMLM. Both preparation methods gave comparable results.

**Long time-lapse 2D SMLM in live cells**. HeLa cells were cultured and plated in phenol red-free growth medium (Gibco). Before imaging, the cells were washed

with PBS and compound PFF-**1** (500 nM) was added 10 min prior to imaging. Read-out was achieved with a laser at 561 nm (0.25 kW cm$^{-2}$, 20 ms) for a total of 100,000 frames. Initial pump acquisition was achieved with a single 405 nm laser pulse (2.6 W cm$^{-2}$, 20 ms). Sequential pump acquisition was achieved with a 405 nm pulse irradiation (2.6 W cm$^{-2}$, 20 ms) repeated every 10 min. An incubator with controllable temperature (37 °C) and $CO_2$ (5%) was used to maintain the cells in an appropriate environment during these long measurements.

**Super-resolution imaging**. HeLa cells were plated in phenol red-free growth medium (Gibco). Cells were washed with PBS (0.5 mL) before imaging and incubated with compound PFF-**1** at the indicated concentration for 10 min. Cells were imaged using a Nikon N-STORM, microscope (Nikon, UK Ltd.) equipped with an SR Apochromat TIRF 100 × 1.49 N.A. oil immersion objective lens. An excitation laser at a wavelength $\lambda = 561$ nm illuminated the sample in HILO mode[40]. Fluorescence was detected with an iXon DU897 (Andor) EM-CCD camera (16 × 16 μm$^2$ pixel size) for 2D SMLM and a Hamamatsu Orca Flash 4 v3 (6.5 × 6.5 μm$^2$ pixel size) for 3D SMLM with an exposure time of 10 ms. An in-built focus-lock system (PFS) was used to prevent axial drift of the sample during data acquisition. The emission was collected and passed through a laser QUAD filter set for TIRF applications (Nikon C-NSTORM QUAD 405/488/561/647) comprising laser clean-up, dichroic and emission filters. The laser excitation at 561 nm had a power density of 0.25 kW cm$^{-2}$. 2D and 3D SMLM camera frames were recorded at 46 or 100 frames s$^{-1}$, respectively and for the later an adaptive optics plug and play accessory MicAO 3DSR for SMLM (Imagine Optic, France) was used. From each image stack, a reconstructed super-resolved image was generated using Thunder-storm (FIJI).

**3D SMLM**. HeLa cells were cultured and plated in phenol red-free growth medium (Gibco). Before imaging, the cells were washed with PBS and compound PFF-**1** (500 nM) was added to the medium 10 min prior to imaging. An adaptive optics plug and play accessory MicAO 3DSR for SMLM (Imagine Optic, France) was used. This module was inserted between the microscope side port and the sCMOS, Hamamatsu Orca Flash 4 v3 camera. As a second step, a controllable and aberration-free astigmatism was introduced to the point spread function (PSF) of single molecules, which allowed us to perform 3D SMLM based on astigmatism by determining the z positions of the fluorescent molecules based on the ellipticity of their PSF[41]. The laser excitation was at 561 nm with a laser excitation intensity of 0.25 kW cm$^{-2}$.

**Single-molecule localization and image analysis**. Molecules were localized using the ThunderSTORM plugin of Fiji (ImageJ)[42]. Signals were detected searching the local intensity maxima in each frame, which were fit using an integrated Gaussian PSF. Localizations with a lateral sigma (FWHM) above 350 nm were filtered out. For measuring the diffusion dynamics, a total of 300 frames were employed and localizations were further filtered by density, applying a distance radius of 50 and 200 nm with a minimum number of 20 and 2 neighbors for 2D and 3D reconstructions, respectively.

**Single-vesicle tracking and analysis**. Single-vesicle tracks were reconstructed and analyzed using MosaicSuite (MOSAIC Group) for ImageJ. Thresholding for 2D was done as follows: a radius of 12 pixels was selected with an absolute intensity value of 0.005 with link range of 4 frames. Diffusion coefficients of fast diffusing acidic vesicles in neurons were calculated manually from the distance between two standing positions with respect to their time frame. Diffusion coefficients for slow diffusing vesicles were obtained from MosaicSuite and 30 vesicles, from 3 different cells, were analyzed for each imaging condition.

## Data availability

The crystallographic data have been deposited at the Cambridge Crystallographic Data Centre under CCDC no. 1844696 (PFF-**1**) and copies can be obtained free of charge from www.ccdc.cam.ac.uk/data_request/cif.
Further information on experimental design is available in the Nature Research Reporting Summary.
Additional data that support the findings of this study are available from the corresponding author upon reasonable request.

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

## Acknowledgments

This work was supported by ETH Zurich (grant ETH-02 16-1). We thank Dr. Nils Trapp for X-ray crystallography, Ms. Jade Nguyen for assistance with cloning, and Dr. Giovanni Bassolino for preliminary experiments. We acknowledge the Scientific Center for Optical and Electron Microscopy (ScopeM, ETH Zurich) for access to microscopy facilities and the Swiss National Supercomputing Center for access to the Euler computer cluster.

## Author contributions

E.A.H. performed all the experiments and analyzed the data. E.A.H. and D.P. performed SMLM experiments and analyzed the data. P.R.-F. designed and supervised the research, and analyzed the data. E.A.H. and P.R.-F. wrote the paper with input from D.P.

## Additional information

**Competing interests:** The authors declare no competing interests.

