## [Peer Review File · Nature Communications]

Reviewers' comments:

Reviewer #1 (Remarks to the Author):

In their manuscript, Rivera-Fuentes and colleagues report on the synthesis and application of the photoactivatable fluxional fluorophore PFF-1. When synthesized, this molecule is in a non-fluorescent conformation. Upon irradiation with UV light it is transferred into another conformation, where it is in equilibrium between a fluorescent and a non-fluorescent state.

The concept of fluxional fluorophores is not new; however, they have not been used previously for super-resolution imaging (to this reviewer's limited knowledge).

The synthesis and application of such a fluorophore warrants publication in Nature Comms. In fact, I am enthusiastic about the concept, as it may open up new strategies for the design of labels for localization based super-resolution microscopy. Hence I support publication.

However, in the current form, I do have very strong concerns on the manuscript as the authors seem to 'oversell' their findings with the consequence that the data do not support the claims:

Major concerns:

1) On various occasions the authors claim to have performed live-cell super-resolution microscopy. Most readers (including this reviewer) would assume that they have recorded cellular structures otherwise blurred by diffraction. I do not see convincing evidence for this and hence the term super-resolution microscopy appears to be misleading. In fact, the manuscript shows nicely that the dye PFF-1 can be localized (in living cells) with high spatial precision using SMLM. This is used for tracking of vesicles, but not for resolving cellular structures. In their images they record only a very small subset of the molecules within these vesicles. To this reviewer's understanding, this is not what most readers would call super-resolution microscopy.

It would be ideal if the authors could use PFF-1 to resolve structures otherwise blurred in a cell. The alternative would be to down tune the manuscript and to precisely describe what has been done.

2) The authors claim that they observe minimal phototoxicity and no apparent photobleaching. I have no doubts about the correctness of their observations. However, the comparison is misleading: Since they do not resolve structures, they do not need to collect the localizations of many fluorophores. Therefore, they can rely on as few as 23 camera frames, whereas typical SMLM relies necessarily on several 10,000 frames. This is just an unfair comparison and this needs clarification.

3) This point is related to 2): In contrast to the authors, this reviewer believes that most current SMLM applications do not rely on photoactivation of dyes and proteins (see introduction of the manuscript). Popular methods such as (d)STORM/GSDIM or PAINT do not require photoactivation. The authors should check their statements and support them with citations. It would benefit the paper to realistically compare the required light intensities for imaging of PFF-1 with data from typical STORM experiments.

Minor point

It would be nice if the authors would comment on the possibility to use PFF-1 to label antibodies or to use it in live cell labeling approaches such as Halo- or SNAP-tag labelling.

Reviewer #2 (Remarks to the Author):

Manuscript by P. Rivera-Fuentes et al. describes the development of self-blinking fluorophore whose fluxionality can be controlled by photoirradiation. This dually controlled fluorescent molecule exhibits the photoisomerization of acylhydrazone, in which the Z-form is at the fast equilibrium

between fluorescently off- and on-states. The molecule is applicable to single-molecule localization microscopy (SMLM) that require only mild photoirradiation. By mapping single molecules after controlled photoactivation of the self-blinking states, they performed time-lapse SMLM imaging of lysosomes with a localization precision of 33 nm over 30 min. The generality of the imaging methodology was confirmed by its application to mitochondria and synaptic vesicle imaging.

I think that the concept is interesting, and the molecule seems to be sufficiently useful to prove the value of this type of SMLM. The synthesis and characterization of compounds were solidly performed.

However, for the fair evaluation of the system, authors should collect/provide more solid data about the basic design of the molecule. Also, I think more explanation/solid data are required for the basis of selective staining of the targets in live cell imaging. Therefore, I cannot recommend the publication of this manuscript as it is. Major revisions with enough new experiments and evidences are needed before accepting to Nat Comm by addressing the following issues.

1. The quantum yield of the photoconversion from E to Z isomer should be provided.
2. Related to #1, Figure 2c is confusing by that the photoconversion efficiency and self-blinking capability are discussed as a whole event. The pH dependency of speed/yield of photoconversion and the pH dependency of fluorescence in each isomer state should be separately measured.
3. In Figure 3a, it might be helpful to show the result with compound 2.
4. In the experiment of synaptic vesicle tracking, the evidence is not fully provided that what author is monitoring is truly "synaptic vesicle". More explanation about the experimental setting and a solid evidence are required.

Reviewer #3 (Remarks to the Author):

The manuscript reports the development of a new spontaneously blinking dye that can be used for single-molecule localization microscopy in living cells. In contrast to other photoactivatable or spontaneously blinking dyes the new dye termed "fluxional molecule" can be photoactivated upon irradiation with light at 405 nm to a form which shows spontaneous blinking between two isomers, a fluorescent and a nonfluorescent form. Hence, the concentration of dyes in the blinking form can be controlled by repeated activation with light at 405 nm and thus enables long lasting single-molecule switching experiments. The potential of the dye for live-cell super-resolution microscopy is demonstrated by adding the dye to HeLa cells and neurons. In both cases the dye penetrates the plasma membrane and accumulates in vesicles, i.e. either lysosomes or synaptic vesicles, which can then, after activation of a subset of dyes upon irradiation at 405 nm, be tracked by super-resolution microscopy.

While I like the idea of fluxional molecules for super-resolution imaging especially the precise control of the concentration of spontaneously blinking molecules I am doubting that the new dye can compete with the already successfully used single-molecule super-resolution microscopy dyes such as Alexa Fluor 647, Cy5 and HMSiR.

In the introduction the authors state that HMSiR performs well only in the low-polarity environment of membranes. I agree that some papers highlight the performance of HMSiR for membrane imaging but the general statement made is actually not true, i.e. HMSiR can also be used successfully for super-resolution imaging of other cellular structures.

Very long time-lapse SMLM experiments without apparent photobleaching and minimal phototoxicity have also been performed with other dyes such as PA-FPs and spontaneously blinking dyes. It is just a matter of definition of phototoxicity. A simple MTT test is for sure not appropriate to claim low phototoxicity of the dye. Have the cells been irradiated or just treated with the dye (100 μ M)? However, the most important concerns I have are:

- In order to demonstrate that the dye can be used successfully for super-resolution imaging of cellular structures the fraction of molecules residing in the fluorescent and non-fluorescent form is crucial. From Fig. 3b it looks like the on/off ratio is too high to enable super-resolution microscopy at high labeling densities. What is the typical on/off ratio of the dye? How long would it take to

resolve a complex structure like the actin skeleton of a cell? A simple experiment labeling actin via phalloidin would answer the question.

- The localization precision is worse than the precisions reported for Alexa 647 or Cy5 in typical STORM experiments. Furthermore, there are photoactivatable dyes on the market that do not require photoactivation and enable imaging with localization precisions of ~ 10 nm (e.g. Michie et al JACS 2017).

- Vesicle tracking with high temporal resolution has been shown by STORM at similar if not higher spatial resolution (Jones et al. Nat Methods 2011).

- The new dye does not exhibit a functional group for specific labeling of molecules of interest. All experiments shown used non-specific labeling of intracellular vesicles by endocytic uptake. As such it remains speculation if the dye can be used successfully for specific labeling of cellular structures and how this modification would change the switching performance of the dye.

Point-by-point Response to Comments by Reviewers

Reviewer 1

General comments. *In their manuscript, Rivera-Fuentes and colleagues report on the synthesis and application of the photoactivatable fluxional fluorophore PFF-1. When synthesized, this molecule is in a non-fluorescent conformation. Upon irradiation with UV light it is transferred into another conformation, where it is in equilibrium between a fluorescent and a non-fluorescent state.*

The concept of fluxional fluorophores is not new; however, they have not been used previously for super-resolution imaging (to this reviewer's limited knowledge). The synthesis and application of such a fluorophore warrants publication in Nature Comms. In fact, I am enthusiastic about the concept, as it may open up new strategies for the design of labels for localization based super-resolution microscopy. Hence I support publication.

However, in the current form, I do have very strong concerns on the manuscript as the authors seem to 'oversell' their findings with the consequence that the data do not support the claims:

Major concerns:

Comment 1. *On various occasions the authors claim to have performed live-cell super-resolution microscopy. Most readers (including this reviewer) would assume that they have recorded cellular structures otherwise blurred by diffraction. I do not see convincing evidence for this and hence the term super-resolution microscopy appears to be misleading. In fact, the manuscript shows nicely that the dye PFF-1 can be localized (in living cells) with high spatial precision using SMLM. This is used for tracking of vesicles, but not for resolving cellular structures. In their images they record only a very small subset of the molecules within these*

vesicles. To this reviewer's understanding, this is not what most readers would call super-resolution microscopy. It would be ideal if the authors could use PFF-1 to resolve structures otherwise blurred in a cell. The alternative would be to down tune the manuscript and to precisely describe what has been done.

Reply 1. We thank the reviewer for her/his positive feedback and are happy to read that she/he is enthusiastic about the concept of our dye. Whereas it is true that we do not image a specific molecular target, we argue that organelles are cellular structures that can be resolved beyond the limit of diffraction without having to label a specific macromolecule. Good examples of papers describing similar experiments include *Proc. Natl. Acad. Sci. U. S. A.* **109**, 13978 (2012) and *Nat. Biotechnol.* **35**, 773 (2017)). To demonstrate that we are also able to distinguish cellular features beyond the limit of diffraction, we have included as Supplementary Fig. 14 some examples of how our dye allows us to image vesicles with super-resolution and provide diffraction-limited images as a comparison. We also show that using our dye we can measure the size of a small synaptic vesicle (33 nm). This value agrees with measurements obtained from electron microscopy (*J. Comp. Neurol.* **514**, 343 (2009)). Furthermore, we demonstrate that we can resolve two larger vesicles (75 and 80 nm) that are separated by only 128 nm (center-to-center distance), all within an area of about 250 x 250 nm. These are clear cases of cellular structures (lysosomes or synaptic vesicles) that we were able to resolve and would otherwise be blurred by diffraction. Finally, it is also worth mentioning that to reconstruct these super-resolved images, we recorded an average density of about 1,100 molecules per μm^2 within vesicles. This molecular density (α) corresponds to a Nyquist-limited resolution ($2\alpha^{-1/2}$) of ~60 nm, which is remarkable considering that these reconstructed snapshots were obtained within 0.5 s. Of course, the Nyquist-limited resolution could be improved at the cost of decreasing the time resolution, but because these vesicles move so fast in living cells, we think that this is a good compromise. Considering these results, we think that it is appropriate to claim that we have performed super-resolution microscopy.

Comment 2. *The authors claim that they observe minimal phototoxicity and no apparent photobleaching. I have no doubts about the correctness of their observations. However, the comparison is misleading: Since they do not resolve structures, they do not need to collect the localizations of many fluorophores. Therefore, they can rely on as few as 23 camera frames, whereas typical SMLM relies necessarily on several 10,000 frames. This is just an unfair comparison and this needs clarification.*

Reply 2. We were able to resolve vesicles (Nyquist-limited resolution of ~60 nm) with as few as 23 frames, which means that we can reconstruct a super-resolved snapshot of vesicles every 23 frames (0.5 s). For our long time-lapse imaging (>30 min), however, we obtained more than 6,500 consecutive reconstructed snapshots, which means that we recorded ~150,000 frames in total. At the end of this very long acquisition, in which cells were exposed to light for more than

30 min, we still detect as many single molecules as in the beginning of the experiment (Figure 4c), which is why we conclude that we have no apparent photobleaching. As a comparison, Jones *et al.* (*Nat. Methods* **8**, 499 (2011)) were also able to reconstruct super-resolved snapshots of vesicles with only a few frames, but their total acquisition was limited to 50 s because of photobleaching of their label (AlexaFluor 647). In contrast, we could image for over 30 min without a substantial loss in single-molecule localizations (Figure 4c). In terms of phototoxicity, we evaluated the fitness of the cells that were imaged (that is, after exposure to ~150,000 frames, >30 min) by analyzing morphological changes, membrane permeability and activation of caspase-3 (early apoptosis), and saw nearly no effect (Supplementary Fig. 13). Based on these three indicators, we conclude that the induced phototoxicity is not substantial at this point.

Comment 3. *This point is related to 2): In contrast to the authors, this reviewer believes that most current SMLM applications do not rely on photoactivation of dyes and proteins (see introduction of the manuscript). Popular methods such as (d)STORM/GSDIM or PAINT do not require photoactivation. The authors should check their statements and support them with citations. It would benefit the paper to realistically compare the required light intensities for imaging of PFF-1 with data from typical STORM experiments.*

Reply 3. We have added a few sentences and references to the main manuscript to clarify this issue. It is true that dSTORM is a popular SMLM implementation, but more often than not, it relies on photoactivation with an activation laser (~405 nm) to provide control over the degree of blinking. For example, the abstract of the original publication of dSTORM (*Angew. Chem. Int. Ed.* **47**, 6172 (2008)) states: “dSTORM uses conventional photoswitchable fluorescent dyes that can be reversibly cycled between a fluorescent and a dark state by irradiation with light of different wavelengths.” The great majority of dSTORM examples rely on photoactivation, for example those using probes that depend on thiol-containing buffers (*J. Am. Chem. Soc.* **131**, 18192 (2009)), direct photoswitching of commercial dyes (*Proc. Natl. Acad. Sci. U. S. A.* **109**, 13978 (2012)), diazoindanone-based fluorophores (*Angew. Chem. Int. Ed.* **49**, 3520 (2010) and *Nat. Methods* **13**, 985 (2016)), and virtually all examples that employ fluorescent proteins (*Chem. Rev.* **117**, 758, (2017)). Some implementations of dSTORM can obtain blinking using a single wavelength (see for example *Nat. Commun.* **9**, 930 (2018)), but at the cost of irradiation powers (2-4 kW cm⁻²) that are about one order of magnitude more intense than in our experiments (0.25 kW cm⁻²). In the revised manuscript we provide, as Supplementary Table SN1, a list of light intensities used in some of these experiments and compare them to our method.

We agree that PAINT does not require photoactivation, but live-cell PAINT imaging of intracellular targets is by no means a routine experiment (a recent example of live-cell PAINT imaging of a membrane-bound target claims to be the first experiment of this kind: *Nat. Methods* **15**, 685 (2018)). Whereas this example is extremely exciting, it definitely does not represent

“most current SMLM applications” and our dye still accomplishes the incredibly challenging task of imaging an intracellular organelle with excellent spatiotemporal resolution, for very long periods of time, in three dimensions, in live, unmodified cells, without using toxic oxygen-scavenging or nucleophile-containing buffers, using only low irradiation powers.

Comment 4. *Minor point*

It would be nice if the authors would comment on the possibility to use PFF-1 to label antibodies or to use it in live cell labeling approaches such as Halo- or SNAP-tag labelling.

Reply 4. As exemplified by our dye MitoPFF-1 (Supplementary Fig. 15), it is possible to conjugate the dye to a targeting vector to label other intracellular organelles. Besides this simple demonstration, we also prepared a taxol derivative of PFF-1, TaxoPFF-1, to image microtubules. With this compound, we sought to show that the photoactivation and fluxionality of PFF-1 are preserved even when it is bound to a large macromolecular target. Under these conditions, we observed excellent photoactivation and thermal switching of TaxoPFF-1 bound to tubulin, allowing us to image microtubules with enhanced resolution (Supplementary Fig. 16). Labeling and washing methods play a very important role in imaging such structures and we are confident that better resolution could be achieved by optimization of these parameters. In our opinion, the most crucial aspect that these experiments demonstrate is that PFF-1 remains fluxional when bound to these macromolecules. Along these lines, we also prepared derivatives of PFF-1 that could be conjugated to either SNAP-tag or HaloTag. Unfortunately, we have not been able to optimize the transfection and labeling conditions to image microtubules in live cells that transiently express tubulin-SNAP-tag or tubulin-HaloTag (data not shown). Once again, this is a matter of optimizing the labeling and washing protocols, and future work will address this crucial aspect of imaging with PFF-1 or derivatives thereof.

Reviewer 2

General comments. *Manuscript by P. Rivera-Fuentes et al. describes the development of self-blinking fluorophore whose fluxionality can be controlled by photoirradiation. This dually controlled fluorescent molecule exhibits the photoisomerization of acylhydrazone, in which the Z-form is at the fast equilibrium between fluorescently off- and on-states. The molecule is applicable to single-molecule localization microscopy (SMLM) that require only mild photoirradiation. By mapping single molecules after controlled photoactivation of the self-blinking states, they performed time-lapse SMLM imaging of lysosomes with a localization precision of 33 nm over 30 min. The generality of the imaging methodology was confirmed by its application to mitochondria and synaptic vesicle imaging.*

I think that the concept is interesting, and the molecule seems to be sufficiently useful to prove the value of this type of SMLM. The synthesis and characterization of compounds were solidly performed.

However, for the fair evaluation of the system, authors should collect/provide more solid data about the basic design of the molecule. Also, I think more explanation/solid data are required for the basis of selective staining of the targets in live cell imaging. Therefore, I cannot recommend the publication of this manuscript as it is. Major revisions with enough new experiments and evidences are needed before accepting to Nat Comm by addressing the following issues.

Comment 1. *The quantum yield of the photoconversion from E to Z isomer should be provided.*

Reply 1. We were able to separate the E and Z isomers by HPLC following photoconversion and measured the quantum yields of photoisomerization accurately at pH 5 and pH 7.4 using this HPLC method. A detailed description of how the experiments were performed was added to the Supplementary Note, key experimental data are depicted in Supplementary Figs. 3–6 and the values of the quantum yields are now discussed in the main manuscript.

Comment 2. *Related to #1, Figure 2c is confusing by that the photoconversion efficiency and self-blinking capability are discussed as a whole event. The pH dependency of speed/yield of photoconversion and the pH dependency of fluorescence in each isomer state should be separately measured*

Reply 2. This experiment was very interesting and we thank the referee for suggesting it. We determined the yield of photoconversion at two pH values and also determined the fraction of fluorescent and non-fluorescent forms for each acylhydrazone isomer at the two relevant pH values. The details of these experiments were added to the Supplementary Note, the values are discussed in the main manuscript, and key data are presented as Supplementary Figs. 6–8.

Comment 3. *In Figure 3a, it might be helpful to show the result with compound 2.*

Reply 3. We added this information, however, as Supplementary Fig. 9.

Comment 4. *In the experiment of synaptic vesicle tracking, the evidence is not fully provided that what author is monitoring is truly “synaptic vesicle”. More explanation about the experimental setting and a solid evidence are required.*

Reply 4. We performed co-localization analysis of our dye with FM1-43 (also known as SynaptoGreen™ C4), a dye that that is widely used to stain recycled synaptic vesicles. The results of these experiments are now mentioned in the main manuscript and were added as

Supplementary Fig. 18. The details of the experimental protocol were added to the Supplementary Note.

Reviewer 3

General comments. *The manuscript reports the development of a new spontaneously blinking dye that can be used for single-molecule localization microscopy in living cells. In contrast to other photoactivatable or spontaneously blinking dyes the new dye termed “fluxional molecule” can be photoactivated upon irradiation with light at 405 nm to a form which shows spontaneous blinking between two isomers, a fluorescent and a nonfluorescent form. Hence, the concentration of dyes in the blinking form can be controlled by repeated activation with light at 405 nm and thus enables long lasting single-molecule switching experiments. The potential of the dye for live-cell super-resolution microscopy is demonstrated by adding the dye to HeLa cells and neurons. In both cases the dye penetrates the plasma membrane and accumulates in vesicles, i.e. either lysosomes or synaptic vesicles, which can then, after activation of a subset of dyes upon irradiation at 405 nm, be tracked by super-resolution microscopy.*

While I like the idea of fluxional molecules for super-resolution imaging especially the precise control of the concentration of spontaneously blinking molecules I am doubting that the new dye can compete with the already successfully used single-molecule super-resolution microscopy dyes such as Alexa Fluor 647, Cy5 and HMSiR.

Comment 1. *In the introduction the authors state that HMSiR performs well only in the low-polarity environment of membranes. I agree that some papers highlight the performance of HMSiR for membrane imaging but the general statement made is actually not true, i.e. HMSiR can also be used successfully for super-resolution imaging of other cellular structures.*

Reply 1. We agree with the referee and amended the text in the main manuscript to be more accurate (see page 3 of the main manuscript)

Comment 2. *Very long time-lapse SMLM experiments without apparent photobleaching and minimal phototoxicity have also been performed with other dyes such as PA-FPs and spontaneously blinking dyes. It is just a matter of definition of phototoxicity. A simple MTT test is for sure not appropriate to claim low phototoxicity of the dye. Have the cells been irradiated or just treated with the dye (100 μ M)?*

Reply 2. We agree that a simple MTT test does not assess phototoxicity. As seen in Supplementary Fig. 13, we evaluated the phototoxicity by comparing a few parameters of the cells before and after a very long time-lapse imaging experiment (30 min, ~150,000 frames). These parameters were changes in morphology (visualized in bright-field images), membrane disruption (assessed by DAPI staining of the nucleus) and activation of caspase-3, a marker of early apoptosis. We did not see significant effects in any of these parameters after very long time-lapse SMLM and that is why we concluded that phototoxicity is minimal after these long time-lapse experiments.

Comment 3. *In order to demonstrate that the dye can be used successfully for super-resolution imaging of cellular structures the fraction of molecules residing in the fluorescent and non-fluorescent form is crucial. From Fig. 3b it looks like the on/off ratio is too high to enable super-resolution microscopy at high labeling densities. What is the typical on/off ratio of the dye? How long would it take to resolve a complex structure like the actin skeleton of a cell? A simple experiment labeling actin via phalloidin would answer the question.*

Reply 3. When compound PFF-1 is in the *E* isomer (before photoactivation) the fraction of fluorescent molecules is very small (~0.004%), as we determined by HPLC and absorbance experiments. In the pure *Z* isomer, the ratio is much higher (39%). Therefore, the on/off ratio in a microscopy experiment depends on how many of the molecules are in the *Z* isomer, which means that the on/off ratio can be ultimately controlled by photoactivation. Lets say, for example, that we irradiate with enough photons to convert 1% of the *E* isomer into the *Z* form. That would give us an on/off ratio of $((99 \times 0.00004) + (1 \times 0.39)) / ((99 \times 0.99996) + (1 \times 0.61)) = 0.394 / 99.606$, which corresponds to a duty cycle of less than 0.004, similar to conventional dyes used for STORM experiments such as ATTO 647N in reducing buffers containing thiols. This is the feature that is innovative about PFF-1, that the on/off ratio depends on photoactivation, but single molecules can be detected based on their fluxional equilibrium.

In Fig. 3b we sought to activate many molecules to measure many blinking events and get good single-particle statistics, but this does not need to be the case. In live-cell imaging experiments, we optimized the incubation times and concentrations, as well as the photoactivation intensity and duration of the pulse, to obtain good labeling density to minimize overlap of single emitters and optimize the Nyquist-limited resolution. Under the conditions reported in the paper, we detected 1,100 molecules per μm^2 within vesicles in 23 camera frames. This molecular density (α) corresponds to a Nyquist-limited resolution ($2\alpha^{-1/2}$) of ~60 nm. We observe some overlap of single emitters, which erodes the localization precision, but we compromised on this aspect in favor of being able to obtain a good time resolution (0.5 s per reconstructed snapshot).

In summary, it is not straightforward to know exactly what is the on/off ratio of the dye within vesicles in a live cell experiment because we do not know how many molecules are converted to the *Z* isomer inside the cell, but optimization of the labeling and photoactivation parameters led to a good compromise between single-molecule localization, temporal resolution and Nyquist-

limited resolution. More importantly, the advantage of PFF-1 and our method in general, is that the user can decide which aspect to optimize (temporal or spatial resolution) simply by tuning the intensity and/or duration of the photoactivation pulse.

Comment 4. *The localization precision is worse than the precisions reported for Alexa 647 or Cy5 in typical STORM experiments. Furthermore, there are photoactivatable dyes on the market that do not require photoactivation and enable imaging with localization precisions of ~ 10 nm (e.g. Michie et al JACS 2017).*

Reply 4. Our typical localization precision (~33 nm or ~11 nm using adaptive optics to correct for aberrations) is certainly “worse” compared with what has been achieved by others employing STORM. However, typical STORM experiments using AlexaFluor 647 or Cy5 derivatives are usually carried out in fixed cells, at much higher illumination intensities, and under special buffers that are generally incompatible with live-cell imaging. Moreover, these experiments rarely suffer from rapid movement of the cellular features that are imaged. In our case, we imaged vesicles that move very rapidly in living cells (with diffusion coefficients up to $3.2 \mu\text{m}^2 \text{s}^{-1}$), and we do so under normal growth medium conditions and with very low laser powers. Regarding the fluorophore reported by Michie *et al.* (*J. Am. Chem. Soc.* **139**, 12406, (2017)), we find it difficult to compare it with our dye because we could not find in either the manuscript or the supporting information the laser intensities that they used. Moreover, it is stated in the main manuscript that SMLM could be carried out without using a UV laser, but in the supporting information, the STORM experimental procedure reads: “A 647 nm laser was used to excite and rapidly switch off the fluorophores. A low amount of 405 nm laser was used to reactivate the dyes throughout the imaging process.” Finally, this paper also reports results of imaging fixed cells in which the target does not move, which helps to improve the localization precision compared with very rapidly moving targets such as lysosomes and synaptic vesicles in living cells.

A relatively more straightforward comparison of our dye can be made against HMSiR, because it has been used in live-cell imaging under biologically relevant conditions. In this case, HMSiR gives a localization precision of 12 nm when the laser power was $4\text{--}10 \text{ kW cm}^{-2}$ (*Nat. Biotechnol.* **35**, 773 (2017)) and 22 nm at 1.5 kW cm^{-2} (*Nat. Chem.* **6**, 681, (2014)). We obtain a localization precision of 33 nm at 0.25 kW cm^{-2} (and a localization precision of 11 nm using adaptive optics under the same conditions). Our irradiation power is at least six times milder than in these reports, yet our localization precision is only 40% worse. This difference in irradiation intensities and the fact that we are imaging fast-moving targets account for the lower localization precision in our experiments. On the other hand, neither AlexaFluor 647 nor HMSiR could have been used for the experiments that we have described because AlexaFluor 647 does not cross the plasma membrane and the fluorescent fraction of HMSiR is too high at low pH for SMLM. We consider that the relatively modest localization precision that we obtained, compared to what can be

measured in fixed samples, is a small price to pay compared to the considerable advantage of performing live-cell experiments under biologically relevant conditions.

Comment 5. *Vesicle tracking with high temporal resolution has been shown by STORM at similar if not higher spatial resolution (Jones et al. Nat Methods 2011).*

Reply 5. It is true that the paper by Jones *et al.* (*Nat. Methods* **8**, 499 (2011)) reports vesicle tracking at higher spatial and similar temporal resolution. However, it is important to notice the differences between this study and ours. Most importantly, they did not track rapidly moving vesicles. As they mention in their paper, they determined a diffusion coefficient for their target (transferrin) of $0.06 \mu\text{m}^2 \text{s}^{-1}$, which is nearly 2 orders of magnitude slower than the fast-moving vesicles that we imaged ($3.2 \mu\text{m}^2 \text{s}^{-1}$). This slower movement of transferrin facilitates the localization of single molecules giving a better localization precision. Moreover, these experiments were carried out under continuous 405 nm irradiation ($1.5\text{--}150 \text{ W cm}^{-2}$) and 561 nm ($5\text{--}20 \text{ kW cm}^{-2}$), whereas in our case only three short (20 ms) pulses of 405 nm irradiation (2 W cm^{-2}) were necessary and imaging was carried out at 561 nm (0.25 kW cm^{-2}). The irradiation power of nearly two orders of magnitude higher at 561 nm certainly accounts for an improvement in the localization precision in the study of Jones *et al.* Importantly, they could only image for about 50 s because of photobleaching, whereas we could image for over 30 min. Once again, we believe that being able to image for much longer times, in live cells, under non-toxic buffer or irradiation conditions outweighs the fact that we “only” obtain localization precisions of about 33 nm (11 nm with adaptive optics).

Comment 6. *The new dye does not exhibit a functional group for specific labeling of molecules of interest. All experiments shown used non-specific labeling of intracellular vesicles by endocytic uptake. As such it remains speculation if the dye can be used successfully for specific labeling of cellular structures and how this modification would change the switching performance of the dye.*

Reply 6. As exemplified by our dye MitoPFF-1 (Supplementary Fig. 15), it is possible to conjugate the dye to a targeting vector to label other intracellular structures. Besides this simple demonstration, we also prepared a taxol derivative of PFF-1, TaxoPFF-1, to image microtubules. With this compound, we sought to show that the photoactivation and fluxionality of PFF-1 are preserved even when it is bound to a large macromolecular target. Under these conditions, we observed excellent photoactivation and thermal switching of TaxoPFF-1 bound to tubulin, allowing us to image microtubules with moderate resolution (Supplementary Fig. 16). Labeling and washing methods play a very important role in imaging such structures and we are confident that better resolution could be achieved by optimization of these parameters. In our opinion, the most crucial aspect that these experiments demonstrate is that PFF-1 remains fluxional when bound to these macromolecules. Along these lines, we also prepared derivatives of PFF-1 that

could be conjugated to either SNAP-tag or HaloTag. Unfortunately, we have not been able to optimize the transfection and labeling conditions to image microtubules in live cells that transiently express tubulin-SNAP-tag or tubulin-HaloTag (data not shown). Once again, this is a matter of optimizing the labeling and washing protocols, and future work will address this crucial aspect of imaging with PFF-1 or derivatives thereof.

Finally, we want to thank all three referees for their constructive feedback, which has greatly improved the quality of this paper.

REVIEWERS' COMMENTS:

Reviewer #1 (Remarks to the Author):

The authors have addressed my previous concerns appropriately. I strongly support publication of this manuscript in Nature Communications.

Reviewer #2 (Remarks to the Author):

I re-reviewed the revised manuscript by Prof. Rivera-Fuentes and colleagues, and found some new interesting data were provided. However, I could not recommend this manuscript to be accepted in Nat Commun from the following reasons.

As I requested the authors, photoconversion and self-blinking capability should be separated to discuss the property of PFF-1 as a SMLM probe. In this revised manuscript, the authors reported that almost all the PFF-1 existed as E-isomer before irradiation of UV pump light, and the E-isomer preferred spiro-closed form. By the irradiation of UV light (254, 375, and 405 nm), photoisomerization of E-isomer proceeded to give Z-isomer, and some of the Z-isomer existed as open-fluorescent form. They newly reported the data in this revised manuscript that the percentage of the Z-isomer molecules which are in the fluorescent open state was 39% (pages 3 and 4, and supplementary figs. 4-6), and I found one of the biggest issues of PFF-1 as a SMLM probe in this high number. I also found another big issue in the new data about the irreversible chemical property of the Z-form back to E-form (page 4 and supplementary fig. 7). Judging from these properties, PFF-1 is no longer a SMLM probe, but in my opinion, should be categorized as a PALM probe, and the authors need to compare the capability (their spatial resolutions and the quality of images) with known caged fluorophores. I could understand the authors' claim that quite small duty cycles were realized with their PFF-1, but this is also true for PALM imaging by adjusting the power of uncaging light. Therefore, unfortunately, I could not find any advantages of PFF-1 in this manuscript over existing self-blinking SMML probes and caged PALM fluorophores.

Reviewer #3 (Remarks to the Author):

The authors carefully addressed all my concerns. Their reply is convincing and supports the claims of the manuscript. Therefore, I support publication of the manuscript in its revised form.

Reviewer 1

General comments. *The authors have addressed my previous concerns appropriately. I strongly support publication of this manuscript in Nature Communications.*

Reply. We thank the reviewer for her/his insightful feedback and for supporting publication of this work.

Reviewer 2

General comments. *I re-reviewed the revised manuscript by Prof. Rivera-Fuentes and colleagues, and found some new interesting data were provided. However, I could not recommend this manuscript to be accepted in Nat Commun from the following reasons. As I requested the authors, photoconversion and self-blinking capability should be separated to discuss the property of PFF-1 as a SMLM probe. In this revised manuscript, the authors reported that almost all the PFF-1 existed as E-isomer before irradiation of UV pump light, and the E-isomer preferred spiro-closed form. By the irradiation of UV light (254, 375, and 405 nm), photoisomerization of E-isomer proceeded to give Z-isomer, and some of the Z-isomer existed as open-fluorescent form. They newly reported the data in this revised manuscript that the percentage of the Z-isomer molecules which are in the fluorescent open state was 39% (pages 3 and 4, and supplementary figs. 4-6), and I found one of the biggest issues of PFF-1 as a SMLM probe in this high number. I also found another big issue in the new data about the irreversible chemical property of the Z-form back to E-form (page 4 and supplementary fig. 7). Judging from these properties, PFF-1 is no longer a SMLM probe, but in my opinion, should be categorized as a PALM probe, and the authors need to compare the capability (their spatial resolutions and the quality of images) with known caged fluorophores. I could understand the authors' claim that*

quite small duty cycles were realized with their PFF-1, but this is also true for PALM imaging by adjusting the power of uncaging light. Therefore, unfortunately, I could not find any advantages of PFF-1 in this manuscript over existing self-blinking SMLM probes and caged PALM fluorophores.

Reply. We thank the reviewer for evaluating our revised manuscript. We agree that for a purely fluxional or “spontaneously blinking” dye a 39% of fluorescent molecules at equilibrium is too high for SMLM. In our case, this fraction corresponds only to the Z isomer, which is obtained from photoactivation. If about 1% of molecules is converted to the Z isomer, only 0.39% of the total molecules would be fluorescent. The difference between our probe and traditional PALM probes, is that the latter are photoactivated to a nearly 100% fraction of fluorescent molecules that do not interconvert thermally and reversibly with a dark isomer. This situation means that in PALM, a low fraction of all molecules is converted into a fluorescent form, which contributes one or very few SMLM frames to reconstruct a super-resolved image. Therefore, another step of photoactivation (usually within one second of acquisition) is needed to convert another small fraction of molecules to a fluorescent state (again nearly 100% fluorescent and non-fluxional). Therefore, multiple photoactivation steps are needed to reconstruct a single super-resolved image. In the case of **PFF-1**, photoconversion gives a fluxional or “spontaneously blinking” dye. In this regard, a single photoactivation pulse populates the fluxional state that provides thousands of SMLM frames, from which many reconstructed images can be obtained, before another photoactivation step is needed (in our case, these photoactivation steps were performed every 10 minutes). This reduced exposure to UV light, which is at least two orders of magnitude smaller than with traditional caged dyes for PALM, decreases phototoxicity allowing us to image live specimens for 30 min with minimal phototoxicity. That is the advantage of photoregulated fluxional fluorophores.

We nonetheless believe that readers should be given a warning about the potential shortcomings of this probe, and therefore added the following paragraph to the conclusions sections of the paper: “Despite its many advantages, PFF-1 also has a few potential shortcomings. For example, the fraction of Z isomers must be kept low to avoid overlap of multiple emitters. Because of this low density of emitting molecules, it would be more difficult to resolve complex cellular structures that require a large number of localizations with high temporal resolution. The development of new photoregulated fluxional fluorophores with increased brightness and blinking speed might ameliorate this situation.”

Reviewer 3

General comments. *The authors carefully addressed all my concerns. Their reply is convincing and supports the claims of the manuscript. Therefore, I support publication of the manuscript in its revised form.*

Reply. We thank the reviewer for her/his insightful feedback and for supporting publication of this work.